# Physiotherapy versus Consecutive Physiotherapy and Cognitive Treatment in People with Parkinson’s Disease: A Pilot Randomized Cross-Over Study

**DOI:** 10.3390/jpm11080687

**Published:** 2021-07-21

**Authors:** Valentina Varalta, Paola Poiese, Serena Recchia, Barbara Montagnana, Cristina Fonte, Mirko Filippetti, Michele Tinazzi, Nicola Smania, Alessandro Picelli

**Affiliations:** 1Neuromotor and Cognitive Rehabilitation Research Center, Section of Physical and Rehabilitation Medicine, Department of Neurosciences, Biomedicine and Movement Sciences, University of Verona, 37134 Verona, Italy; valentina.varalta@univr.it (V.V.); cristina.fonte@univr.it (C.F.); mirko.filippetti@univr.it (M.F.); alessandro.picelli@univr.it (A.P.); 2Neurorehabilitation Unit, University Hospital of Verona, 37126 Verona, Italy; 3Centro Polifunzionale Don Calabria, 37138 Verona, Italy; paola.poiese@centrodoncalabria.it (P.P.); serena.recchia@centrodoncalabria.it (S.R.); barbara.montagnana@centrodoncalabria.it (B.M.); 4Neurology Unit, Movement Disorders Division, Department of Neurosciences, Biomedicine and Movement Sciences, University of Verona, 37134 Verona, Italy; michele.tinazzi@univr.it

**Keywords:** rehabilitation, cognition, movement disorders

## Abstract

Background: Parkinson’s disease (PD) is characterized by motor and cognitive dysfunctions that can usually be treated by physiotherapy or cognitive training, respectively. The effects of consecutive physiotherapy and cognitive rehabilitation programs on PD deficits are less investigated. Objective: We investigated the effects of 3 months of physiotherapy (physiotherapy treatment group) or consecutive physiotherapy and cognitive (physiotherapy and cognitive treatment group) rehabilitation programs on cognitive, motor, and psychological aspects in 20 PD patients. Methods: The two groups switched programs and continued rehabilitation for another 3 months. The outcomes were score improvement on cognitive (Montreal Cognitive Assessment, Frontal Assessment Battery, Trail Making Test, Verbal Phonemic Fluency, Digit Span, and Rey Auditory Verbal Learning), motor (Unified Parkinson’s Disease Rating Scale-III, Berg Balance Scale, Two-Minute Walking Test, and Time Up and Go), and psychological (Beck Depression Inventory and State-Trait Anxiety Inventory) scales. Results: Between-group comparison revealed a significant difference in functional mobility between the two rehabilitation programs. Improvements in walking abilities were noted after both interventions, but only the patients treated with consecutive training showed better performance on functional mobility and memory tasks. Conclusion: Our findings support the hypothesis that consecutive physiotherapy plus cognitive rehabilitation may have a greater benefit than physiotherapy alone in patients with PD.

## 1. Introduction

Parkinson’s disease (PD) is a neurodegenerative disorder characterized by motor and nonmotor symptoms [1]. Motor symptoms include bradykinesia, resting tremor, and postural instability [2], resulting in impaired gait and balance [3]. Common nonmotor manifestations are cognitive deficits in attention, memory, visuospatial, and executive functions [4], and mood disorders, particularly depression and anxiety [5].

While motor manifestations have long been investigated, nonmotor symptoms are now recognized as the main components that interfere with functionality in PD [6]. Cognitive deficits seem to be related to motor abilities [7,8]. Basal ganglia functional connectivity, which is compromised in PD by dopaminergic reduction, may have an important role in modulating both cognitive and motor functions [9].

These findings suggest the importance of evaluating and treating both motor and cognitive dysfunctions. Nonetheless, rehabilitation programs are often limited to physical training. While the effectiveness of physiotherapy in improving motor abilities is well documented [10,11,12,13], it is only in recent years that its effects on cognition have been better investigated [12,13]. Few studies to date have investigated cognitive outcomes before and after motor treatment [12,13]. The improvement in attention and executive functions after physical activity [12,13] suggested a relationship between cognitive and motor improvement in PD in which a common mechanism may underlie cognition and movement [13].

The effects of other rehabilitation interventions besides physiotherapy on motor and cognitive dysfunctions have been investigated in PD patients. An increase in motor and cognitive performance was found after combined (dual-tasking) [14] and consecutive training physiotherapy and cognitive treatment [14,15]. The studies concluded that the effectiveness of combined or consecutive treatment was comparable to motor treatment alone [14,15]. One study suggested that isolated cognitive training may reduce the severity of freezing of gait in PD patients [16]. 

Study findings suggest that physiotherapy [12,13], cognitive [17] combined [14] or consecutive physiotherapy, and cognitive training [14,15] can improve dysfunctions in PD patients. However, usually the studies measured changes in some functional features (motor, cognitive, psychological) but did not deeply assess cognitive or motor abilities. 

The aim of the present study was to investigate the effects of physiotherapy alone versus consecutive physiotherapy and cognitive treatment on cognitive, motor, and psychological aspects in patients with mild to moderate PD.

## 2. Materials and Methods

### 2.1. Study Design

Block randomization for this pilot randomized cross-over trial was generated by a computer with a web tool (randomization.com; accessed on: 25 august 2016). The patients took part in two rehabilitation programs (PT or PCT); each program consisted of 3 months of treatment, and 8 sessions/month. The two groups switched to the PT and the PCT arm, respectively, after a 1-month wash out period (Figure 1).

### 2.2. Ethical Aspects

All participants were outpatients and gave their informed, written consent to participate. The study was carried out according to the Declaration of Helsinki and approved by the Centro Polifunzionale Don Calabria Review Board (no. 05/2016). The patients enrolled in this study are a subgroup involved in a clinical trial registered at http://clinicaltrials.gov (NCT03741959).

### 2.3. Subjects

The study population was composed of 20 patients with confirmed diagnosis of idiopathic PD, according to the Movement Disorder Society (MDS) PD diagnostic criteria [18], and with Hoehn and Yahr stage 3, determined in the “on” phase [19]. Hoehn and Yahr stage 3 was chosen in line with the physiotherapy program, which focused primarily on walking and balance (see Intervention and Procedures subsection for details) and with outcome measures that require sufficient walking ability (2-Minute Walking Test and Time Up and Go).

Exclusion criteria were severe unpredictable “on-off” fluctuations that could have compromised participation in the rehabilitation program, history of alcohol or drug abuse, psychotic disorders, vestibular disorders or paroxysmal vertigo, and other neurological or orthopedic conditions involving the lower limbs (e.g., musculoskeletal diseases, severe osteoarthritis, and peripheral neuropathy). 

All patients received physiotherapy training (PT) or consecutive physiotherapy and cognitive training (PCT). Ten were allocated to the same training in reverse order (Figure 1). Patients were monitored between September 2016 and May 2017. At the end of the study, PT and PCT included 20 patients in each group.

### 2.4. Intervention and Procedures

Patients received group treatment for 50 min/day, 2 days/week, for a total of 24 treatment sessions. Each group was composed of 10 patients with comparable motor and cognitive abilities. The PT program consisted of 24 physiotherapy sessions: one 50 min session/day, 2 days a week, for 12 consecutive weeks. The PCT program consisted of 12 physiotherapy sessions plus 12 cognitive sessions: a 50 min physiotherapy session one day/week and a 50 min cognitive session on another day during the same week for 12 consecutive weeks. Participation in other types of rehabilitation was not permitted during the study period.

The motor interventions were conducted in a well-lit and wide gym by two trained physiotherapists with experience in neuromotor rehabilitation. Each session consisted of three parts with a 5 min rest in between. In adherence to clinical practice guidelines for physical therapy in PD [20], the motor intervention focused on balance, gait, transfers, posture, and upper limb ability. Motor interventions primarily train gait and balance functions because these are usually impaired in PD and have an important impact on daily life activities.

First, patients performed active joint mobilization of the lower limbs (hip, knee, ankle) for 10 min. Lower limb mobilization was carried out with the patient supine (internal/external hip mobilization, active straight leg raise, knee flexion/extension, ankle mobilization) and prone (active hip extension) positions. During each training session, a total of five exercises were performed (four in supine and one in prone position).

Second, patients performed conventional gait and balance therapy for 20 min based on the proprioceptive neuromuscular facilitation concept and aimed at improving both feedforward and feedback postural reactions [21]. Two types of exercises were carried out. In the first of the exercises, patients performed voluntary motor actions in static or dynamic conditions (transferring body weight onto the tips of the toes and onto the heels; bouncing a ball during gait with the two hands alternating to the right and the left side). The second type of exercises trained coordination between leg and arm movement during walking and locomotor dexterity over an obstacle course. During each treatment session, a total of four exercises were performed (two from the first group and two from the second). Each single exercise was repeated three times in 5 min. 

Finally, upper limb, lower limb, and trunk motor coordination exercises were executed for 10 min. Specifically a sequence of two exercises using single-leg stance and a sequence of two exercises associated with upper limb movement were conducted. Each single exercise was repeated two times in 2–3 min.

The physiotherapists assisted patients by demonstrating the exercises and providing verbal instructions.

The cognitive interventions were conducted in a well-lit room by a psychologist with experience in neuropsychological field and cognitive rehabilitation. The objective was to improve cognitive skills by acquiring restorative and compensatory techniques (e.g., memory strategies). Each session consisted of four parts with a 5 min rest in between. To start, the psychologist introduced the aim of the session, then oral and paper-pencil exercises of three cognitive functions were performed in each session. Each function was trained for 10 min followed by a 5 min rest. Memory, concentration, orientation, calculation, dual tasking, and cognitive flexibility were practiced. During the session, the patient stayed seated near a table.

The psychologist supported patients by providing verbal instructions and suggesting useful cognitive strategies.

The TIDieR (Template for Intervention Description and Replication) Checklist are present in the Appendix A.

### 2.5. Data Collection and Assessment Procedures

During the study, patients took their regular PD medications. All underwent cognitive, motor, and psychological assessment during the “on” phase (1 to 2.5 h after having taken their morning dose). The same raters, blinded to the rehabilitation program, evaluated all patients (P.P. performed cognitive and psychological assessment; S.R. performed motor assessment) in an outpatient clinical setting. Motor, cognitive, and psychological assessments were conducted before and after the completion of each rehabilitation program.

### 2.6. Outcome Measures

#### 2.6.1. Primary Outcome Measures

The primary outcome measures were the Montreal Cognitive Assessment (MoCA) [22] and Unified Parkinson’s Disease Rating Scale Part III (UPDRS III) [23].

The MoCA was used to investigate a patient’s global cognitive level. The total score is the sum of all trials, with a maximum score of 30 (best performance) [22]. To decrease possible learning effects between consecutive assessment timing, we used different versions of the MoCA (7.1, 7.2, 7.3) [24].

The UPDRS III was used to assess movement capacity. It consists of 14 items (each rated on a scale from 0 to 4 points) about tremor, slowness (bradykinesia), stiffness (rigidity), and balance. The total score is the sum of all items; the range is from 0 (best performance) to 56 (worst performance) [23].

#### 2.6.2. Secondary Outcome Measures

The secondary outcome measures were other cognitive (Frontal Assessment Battery—FAB-it, Trail Making Test—TMT, F-A-S Verbal Phonemic Fluency Test—FAS, Digit Span Forward—DSF and Backward—DSB, and Rey Auditory Verbal Learning Test—RAVL), psychological (Beck Depression Inventory—BDI and State-Trait Anxiety Inventory—STAI), and motor (Berg Balance Scale—BBS, 2-Minute Walking Test—2MWT, and Time Up and Go—TUG) tests.

The FAB-it assesses executive functions (conceptualization, mental flexibility, programming, sensitivity interference, inhibitory control, and environmental autonomy). It consists of six tests, each of which is rated on a scale from 0 to 3 points. The total score is the sum of all items; the range is from 0 (worst performance) to 18 (best performance) [25].

Selective attention, psychomotor speed, and sequencing skills were evaluated with the TMT part A. The ability to switch attention between two rules and cognitive flexibility were assessed with the TMT part B. The time taken to complete the trials was recorded (longer = worse performance) [26].

The FAS assesses verbal fluency by determining the number of words beginning with a letter (F, A, or S) generated in 60 s. The total score is the average number of words produced (greater = better performance) [27].

Short-term memory was assessed with the DSF. Subjects are asked to repeat forward a list of single digit numbers in the correct order immediately after presentation. The maximum score is 9 (best performance) [28].

Working memory was assessed with the DSB. Subjects are asked to repeat backward a list of single-digit numbers in the correct order immediately after presentation. The maximum score is 8 (best performance) [28].

In order to assess learning and long-term verbal memory abilities, we used the RAVL. The test consists of two parts: immediate recall (RAVL-I) for learning and delayed recall (RAVL-D) for long-term memory. The maximum score for RAVL-I is 75 (best performance) and 15 (best performance) for RAVL-D [29].

The BDI consists of 21 items rated on a 4-point scale of severity of psychological aspects of depression. The total score is the sum of all items; the maximum score is 63 (worst mood) [30].

State anxiety level was assessed with the STAI-Y2, which consists of 20 questions, each rated on a 4-point Likert-like scale. Higher scores are positively correlated with higher levels of anxiety [31].

The BBS is a 14-item scale (each rated on a scale from 0 to 4 points) that evaluates balance abilities during sitting, standing, and positional changes. The total score is the sum of all items; the range is from 0 (worst performance) to 56 (best performance) [32].

The 2MWT measures self-paced walking ability. The total score is the distance (meters) covered in 2 min (greater = better performance) [33].

The TUG is a functional mobility test associated with balance problems and falls in older adults, in which a subject must stand up, walk 3 m, turn around, walk back, and sit down. The time (seconds) taken to complete the test is recorded (longer = worse performance) and it is correlated with the level of functional mobility [34]. Patients can perform the TUG also under two dual-task conditions, one in which they are asked to count backwards from a randomly selected number between 20 and 100 (TUG-COG) and one in which they try to hold a full cup of water steady while walking (TUG-MOT). The time (seconds) taken to complete the test is recorded (longer = worse performance) and is correlated with the level of functional mobility under the dual-task condition [35].

### 2.7. Statistical Analysis

Data were analyzed using IBM SPSS software version 26.0 for Macintosh (IBM Corp., Armonk, NY, USA). Normal distribution of data was determined using the Kolmogorov–Smirnov and Shapiro–Wilk tests, and the homogeneity of variance was assessed with the Levene test. The normal and homogeneous variables were analyzed with two-way mixed ANOVA with a between-individual factor “group” (PT and PCT) and a within-individual factor “time” (pre- and post-treatment). Post hoc comparisons were corrected with the LSD method.

The other variables were analyzed with the Mann–Whitney U-test to compare the effects of treatment between the two groups and with the Wilcoxon signed-rank test for within-group comparison. Descriptive analysis was used to evaluate the effect size measures between groups (Cohen’s d calculation) and the 95% confidence intervals [36]. The alpha level for significance was set at *p* < 0.05.

## 3. Results

For this pilot study, the study population was 20 patients (12 females, 8 males; mean age 70.8 ± 5.09; mean years of schooling 10.15 ± 4.69) with idiopathic PD (mean disease duration 7 ± 3.83 years) recruited from among 33 outpatients consecutively admitted to the Centro Polifunzionale Don Calabria of Verona, Italy, between June and August 2016. No drop-outs or adverse events were recorded during the study. Figure 1 illustrates the study flow diagram.

### 3.1. Baseline

Among outcome measures, FAS, RAVL-I, RAVL-D, and BDI scores resulted as normally distributed (Kolmogorov–Smirnov and Shapiro–Wilk tests, *p* > 0.05) and homogeneous (Levene test, *p* > 0.05).

There were no statistically significant differences in primary (MoCA, p=0.346; UPDRS III, *p* = 0.724) and secondary outcome measures (FAB-it, *p* = 0.955; TMT-A, *p* = 0.655; TMT-B, *p* = 0.891; FAS, *p* = 0.557; DSF, *p* = 0.319; DSB, *p* = 0.931; RAVL-I, *p* = 0.686; RAVL-D, *p* = 0.602; BDI, *p* = 0.666; STAI, *p* = 0.776; BBS, *p* = 0.924; 2MWT, *p* = 0.516; TUG, *p* = 0.482; TUG-COG, *p* = 0.402; TUG-MOT, *p* = 0.198) between the PT and the PCT group before treatment.

### 3.2. Primary Outcomes

Analysis revealed no statistically significant differences between the PCT and the PT group after treatment (MoCA, *p* = 0.257, *z* = −1.133; UPDRS III, *p* = 0.724, *z* = −0,352). Within-group comparison showed significant changes in the UPDRS III scores pre-treatment versus post-treatment for both groups (PCT, *p* = 0.002, *z* = −3,061; PT, *p* = 0.004, *z* = −2.892).

### 3.3. Secondary Outcomes

For the outcome measures analyzed with non-parametric tests, the between-group comparison showed statistically significant differences in TUG (*p* = 0.047; *z* = −1.988) and TUG-MOT (*p* = 0.023; *z* = −2.272) between the PCT and the PT group after treatment. The within-group comparison showed significant changes in pre-treatment versus post-treatment scores for the PCT group in the 2MWT (*p* = 0.006; *z* = −2.726), TUG (*p* = 0.033; *z* = −2.128), and TUG-MOT (*p* = 0.007; *z* = −2.700) and for the PT group in 2MWT (*p* = 0.011; *z* = −2.558).

Table 1 presents the group data and results of the within-group comparison for outcome measures analyzed with non-parametric tests 

For the outcome measures analyzed with parametric tests, ANOVA revealed a principal significant effect of “time” for RAVL-I (F_(1,38)_ 9.459, *p* = 0.004, *η* = 0.199) and RAVL-D (F_(1,38)_ 13.671, *p* = 0.001, *η* = 0.265). The RAVL-I score was significantly higher after treatment (mean score post-treatment RAVL-I 38.15 ± 9.102) compared to before treatment (mean score post-treatment RAVL-I 34.65 ± 8.438). The RAVL-D score was significantly higher after treatment than before treatment (mean score post-treatment RAVL-D 7.5 ± 3.138—mean score pre-treatment RAVL-D 5.85 ± 2.975). Post hoc comparisons revealed significantly higher scores at post-treatment with respect to pre-treatment for RAVL-I and RAVL-D only in the PCT group (mean RAVL-I pre-treatment 34.1 ± 6.813; mean RAVL-I post-treatment 39.4 ± 10.58; mean RAVL-D pre-treatment 5.6 ± 2.854; mean RAVL-D post-treatment 7.65 ± 3.297).

ANOVA showed no group effect of FAS (F_(1,38)_ 0.042, *p* = 0.838, *η* = 0.001), RAVL-I (F_(1,38)_ 0.075, *p* = 0.785, *η* = 0.002), RAVL-D (F_(1,38)_ 0.013, *p* = 0.909, *η* = 0), and BDI (F_(1,37)_ 0.001, *p* = 0.969, *η* = 0). Additionally, no effect of “timeXgroup” interaction was found for FAS (F_(1,38)_ 0.092, *p* = 0.343, *η* = 0.024), RAVL-I (F_(1,38)_ 2.502, *p* = 0.122, *η* = 0.062), RAVL-D (F_(1,38)_ 0.803, *p* = 0.376, *η* = 0.021), and BDI (F_(1,37)_ 0.8091, *p* = 0.374, *η* = 0.21).

Table 2 presents the group data and results for outcome measures analyzed with the parametric tests.

## 4. Discussion

In this pilot randomized cross-over trial, we compared improvement in cognitive, motor, and psychological domain scores after physiotherapy training alone and after consecutive physiotherapy and cognitive treatment in patients with PD.

We observed a statistically significant difference in motor abilities between the two groups after treatment.

There was a change in functional mobility performance, as evaluated with the single- (TUG) and the dual-task condition (TUG-MOT) after PCT. Patients showed an improvement in gait velocity, especially in the dual-task condition (about 4 s faster). In line with a previous study involving other types of patients [37], we speculate that this change may translate into a clinically meaningful improvement for the patients. Our data are shared by the previous study, in which patients who received training with separated gait and cognitive exercises showed an improvement on dual-tasking abilities after treatment [14].

Since the published data suggest a significant correlation between functional mobility (single- or dual-task condition) and cognitive abilities [7], we assume that specific training including cognitive rehabilitation is necessary to also enhance functional mobility under the dual-task condition in PD patients. In our patients, this improvement did not seem possible with physiotherapy alone. We speculate that consecutive treatment provides for training attention and dual-task abilities. This view is shared by previous work that reported a relationship between functional capacity and mobility and severity of cognitive impairment in neurological patients [38].

Our findings for cognitive change indicate that consecutive physiotherapy and cognitive treatment, but not physiotherapy alone, may help to improve performance on delay recall memory ability (as measured with the RAVL-D). This observation is shared by previous studies that reported improvements after cognitive [17] and motor plus cognitive [14,15] training. Furthermore, our data indicate that only the PD patients who underwent this training showed significant improvement in verbal learning (as measured with the RAVL-I). This suggests that cognitive rehabilitation enables patients to improve their memory learning abilities. Specific memory training is needed to increase verbal learning in which correct memory strategies can be acquired.

No changes in cognitive performance were found after treatment in the PT group. Differently from previous studies [13,15,39,40], our data suggest that physiotherapy alone does not result in improved cognitive abilities in PD. We assumed that group treatment might reduce cognitive engagement during rehabilitation exercises. Furthermore, our physiotherapy program, unlike others [39,40], did not include aerobic exercises, which are suggested to induce improvement in brain functional connectivity in the frontal areas [41] that are essential for cognitive functionality in PD patients.

As expected, both training modalities had a positive effect on walking ability (as measured with the 2MWT). This is in line with previous studies that indicated an improvement in motor performance after motor rehabilitation [10,12,39]. However, differently from others [15,39], we found a decrease in global motor performance (as measured with UPDRS III) after both types of training. Unlike other study designs, cross-over studies have natural progression of the disease as a bias. The UPDRS motor score progressed over time, with an annual score increase of 3.3 points [42]. This is in line with the changes in scores in our sample (about 2 points on UPDRS III for both groups during the 8-month study period). However, the pre-post treatment change in score did not have a clinical impact on the PD patients [43]. Due to an unfortunate mistake in the first version of this study, we did not use the revised version of the UPDRS (MDS-UPDRS). Future study protocols will include this version to confirm our data for the UPDRS III.

We believe that the discordant results between the 2MWT and the UPDRS III could stem from the type of intervention. For this study, the focus of the physiotherapy training was primarily on walking and balance, whereas the UPDRS III is a global measure that investigates all movement dysfunctions in patients with PD.

Differently from other studies, ours examined the effects of consecutive physiotherapy and cognitive rehabilitation on different motor abilities (balance, walking, and functional mobility). Barboza et al. investigated motor performance with only one motor outcome (UPDRS III) [15] and Strowen at al.’s study [14] specifically assessed gait velocity under single- and dual-task conditions. By using different outcome measures, we were able to investigate the different motor dysfunctions more deeply.

It is difficult to compare our findings with published data because of the differences in our consecutive training protocol compared to previous studies [14,15]. In our protocol, cognitive and physical training sessions were performed on two separate days (not in the same session) and the patients underwent group (not individual) treatment.

Again, different from a previous study [39], we found stabilization of pre-post treatment of mood (as measured with the BDI), as reported in another study [40]. Summarizing, few studies on rehabilitation in PD include psychological outcome measures, and current knowledge in this field is scarce. A future area of focus is the effect of rehabilitation (cognitive, motor, or combined) on mood in PD patients.

The present study has some limitations. The sample size was small and no follow-up assessment was performed. These limitations reflect the rehabilitation context where the study was conducted. The medical center can accept few patients at one time for rehabilitation care, and following the inclusion criteria, we further reduced the number of patients potentially eligible for the study. Since the majority of the study subjects went on to participate in other training sessions at other clinics immediately after the end of the present study, we could not perform follow-up assessments and compare the effects of our treatment with those of a control group.

In addition, since the patients were not tested in “off” medication, no conclusions can be drawn about the unmedicated state. Another area of focus would be to investigate the effects of cognitive training alone on more than one motor dysfunction in patients with PD and to compare its effectiveness with other approaches (physiotherapy or physiotherapy and cognitive treatment). It might also be interesting to conduct a randomized trial with a non-cross-over design involving two separate patient groups.

Overall, our data indicate a statistical difference pre-post treatment for some outcome measures, suggesting the benefit of consecutive motor and cognitive treatment for functional mobility and long-term memory in PD patients. Nonetheless, we cannot be certain that such differences translate into a clinical change. Further studies are needed to confirm our preliminary results and better investigate the clinical impact of rehabilitation programs in PD patients.

## 5. Conclusions

Our findings suggest potential effects of consecutive motor and cognitive interventions in PD. Our findings may also serve as starting points to better investigate the effects of this rehabilitation program on cognitive, motor, and psychological symptoms in PD patients. Finally, the clinical implications are that the identification and treatment of cognitive deficits are important in patients with PD and that a benefit can be gained with rehabilitation programs that include both motor and cognitive training.

## Figures and Tables

**Figure 1 jpm-11-00687-f001:**
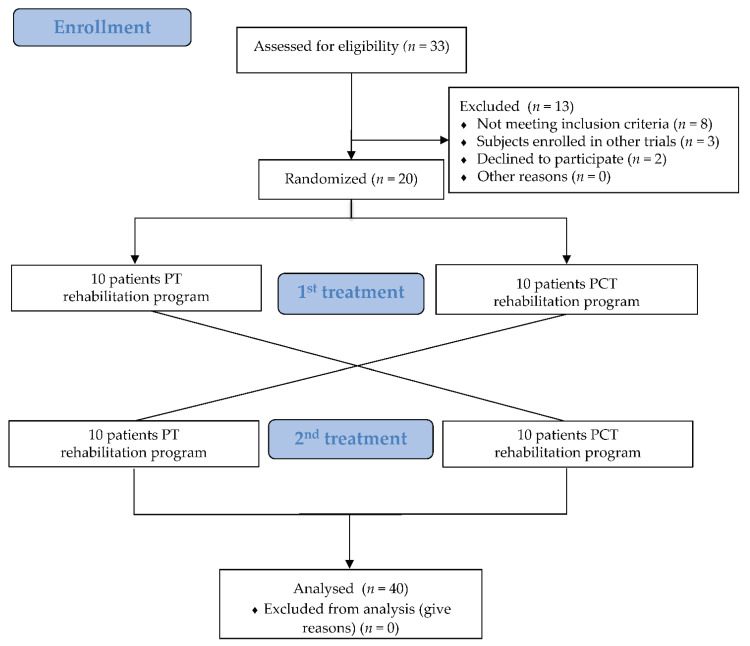
Study flow.

**Table 1 jpm-11-00687-t001:** Within-group comparisons for outcome measures analyzed with non-parametric tests.

Outcome	Rehabilitation Program	Pre Treatment	Post Treatment	Within Group Comparison
				Post vs. Pre-Treatment *p* Value (95% CI)
MoCA (0–30) median (IQR)	PCT	26.5 (24.75; 28)	26 (24; 27.25)	0.099 (−1.41; 0.21)
	PT	26.5 (23; 27)	26 (24.5; 27.25)	0.536 (−1.01; 0.51)
UPDRS-III (0–56) median (IQR)	PCT	9.5 (5; 16.25)	13 (11; 21.5)	0.002 (1.64; 7.76) *
	PT	11 (7.5; 13)	13 (11; 17)	0.004 (1.19; 7.11) *
FABit (0–18) median (IQR)	PCT	16 (15; 18)	16 (13.5; 17)	0.243 (1.99; 0.59)
	PT	17 (14.75; 18)	17.5 (13.5; 18)	0.893 (−1.61; 1.01)
TMTa (seconds) mean (SD)	PCT	57.55 (24.37)	59.65 (37.21)	0.808 (−11.86; 16.06)
	PT	55.6 (25.37)	56.2 (27.77)	0.872 (−6.90; 8.10)
TMTb (seconds) mean (SD)	PCT	197.4 (93.54)	200.6 (92.96)	0.73 (−15.88; 22.28)
	PT	195.1 (94.27)	197.4 (94.07)	0.977 (−20.72; 25.32)
DSF (0–9) median (IQR)	PCT	6 (5; 6)	6 (5; 6)	0.564 (−0.47; 0.27)
	PT	5 (5; 6)	6 (5;6)	0.109 (−0.08; 0.68)
DSB (0–8) median (IQR)	PCT	4 (3; 4.25)	4 (4; 4)	0.571 (−0.40; 0.70)
	PT	4 (3; 4)	4 (3.75; 5)	0.35 (−0.35; 0.95)
BBS (0–56) median (IQR)	PCT	53.5 (50; 55)	54 (48; 55)	0.954 (2.37; 1.37)
	PT	52 (50.5; 55)	54 (50.75; 55)	0.652 (−3.62; 1.02)
2MWT (meters) mean (SD)	PCT	107.28 (30.54)	130.53 (38.32)	0.006 (7.63; 38.88) *
	PT	112.9 (48.48)	130.6 (39.07)	0.011 (5.13; 30.23) *
TUG (seconds) mean (SD)	PCT	11.96 (6.65)	10.89 (6.85)	0.033 (−2.37; 1.37) *
	PT	10.49 (4.46)	11.12 (6.73)	0.823 (−0.73; 1.98)
TUG-COG (seconds) mean (SD)	PCT	14.30 (6.99)	12.85 (6.96)	0.067 (−2.93; 0.05)
	PT	12.58 (5.19)	12.66 (7.37)	0.601 (−1.38; 1.54)
TUG-MOT (seconds) mean (SD)	PCT	13.37 (6.53)	9.36 (2.23)	0.007 (−9.05; -0.86) *
	PT	10.99 (4.98)	11.6 (6.94)	0.149 (−2.84; 5.11)
STAI-Y2 (0–80) median (IQR)	PCT	43 (38.75; 50)	43 (36; 48.25)	0.432 (−5.18; 3.18)
	PT	41 (36; 49)	40 (35; 50.25)	0.904 (−4.79; 6.79)

Abbreviations: IQR = interquartile range; SD = standard deviation; CI = confidence interval; MoCA = Montreal Cognitive Assessment; UPDRS-III = Unified Parkinson’s Disease Rating Scale part III; FAB-it = Frontal Assessment Battery-Italian version; TMT = trail making test; DSF = digit Span Forward; DSB = Digit Span Backward; BBS = Berg Balance Scale; 2MWT = 2 min walking test; TUG = Time Up and Go; TUG-COG = cognitive; TUG-MOT = motor; STAI-Y2 = State-Trait Anxiety Inventory part Y2; * = statistically significant (*p* < 0.05).

**Table 2 jpm-11-00687-t002:** Group data and results for outcome measures analyzed with parametric tests.

Outcome	Rehabilitation Program	Pre Treatment	Post Treatment	Repeated Measures ANOVA		Post Hoc Analysis
				Group Between-Subjects	Time Whitin-Subjects	Whitin-Group
				*p*	*p*	Post vs. Pre-Treatment *p* Value (95% CI) *p*
FAS (no. words) mean (SD)	PCT	11.65 (3.99)	13.02 (3.98)	0.838	0.075	/
	PT	12.39 (3.91)	12.82 (5.57)			/
RAVL-I (0-75) mean (SD)	PCT	34.1 (6.81)	39.4 (10.58)	0.785	0.004 *	0.002 (2.04; 8.56) *
	PT	35.2 (9.96)	36.9 (7.4)			0.297 (−1,56; 4.96)
RAVL-D (0-15) mean (SD)	PCT	5.6 (2.86)	7.65 (3.3)	0.909	0.001 *	0.002 (0.77; 3.33) *
	PT	6.1 (3.14)	7.35 (3.05)			0.055 (−0.03; 2.53)
BDI (0-63) mean (SD)	PCT	12.37 (7.03)	12 (7.61)	0.906	0.577	/
	PT	11.3 (8.23)	12.9 (7.52)			/

Abbreviations: SD = standard deviation; CI = confidence interval; FAS = fonemic verbal fluency; RAVL = Rey Auditory Verbal Learning I = immediate, D = delay; BDI = Beck Depression Inventory; * = statistically significant (*p* < 0.05).

## Data Availability

Data is contained within the article or Appendix A.

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
