# Peer review of "Physiotherapy versus Consecutive Physiotherapy and Cognitive Treatment in People with Parkinson’s Disease: A Pilot Randomized Cross-Over Study"

_jpm, 2021, doi:10.3390/jpm11080687_

Round 1
Reviewer 1 Report
Dear Srs.
I carefully read this manuscript and I still have some comments:
Comment 1
Please review the English throughout the text. Grammatical erros are constant and make it difficult to follow.
Eg Abstract: Parkinson’s disease (PD) is characterized by motor and cognitive dysfunctions that usually be treated respectively by physiotherapy or cognitive training.
Should be: Parkinson’s disease (PD) is characterized by motor and cognitive dysfunctions that are usually treated by physiotherapy or cognitive training respectively.
Comment 2
The abstract needs significant improvement to be simpler and easy to follow. Understanding what “complete rehabilitation program” means is very unclear.
Comment 3
Overall Background is very dense and not easy to read.
Additionally more references are welcomed for sentences such as:
While the effectiveness of physiotherapy in improving motor abilities is well documented [11, MORE,…], it is only in recent years that its effects on cognition have been better investigated (REF MISSING).
Comment 4
If you are applying a rehabilitation protocol assessing general outcomes, why was only Hoehn and Yahr stage 3 included?
Comment 5
The motor intervention selected as comparison aimed at improving what? Your description doesn’t justify why and what for? Is it the aerobic component you aiming for so that you would see more cognitive benefits? (according to the literature cited in the background).
Description made: First, patients performed active joint mobilization of the lower limbs (hip, knee, and ankle) for 10 minutes (5 minutes supine, 5 minutes prone position). Second, they performed conventional gait and balance therapy based on the proprioceptive neuromuscular facilitation concept for 20 minutes. Finally, upper limb, lower limb, and trunk motor coordination exercises were executed for 10 minutes.
In order to increase reproducibility of this study, a clear description of the interventions applied is required with justification with evidenced based principles.
Comment 6
There is a need to standardize wording, often used; physiotherapy isolated…vs physiotherapy training alone vs only consecutive physiotherapy vs cognitive “pure” training.
Comment 7
Discussion section should be more modest and objective. Too many assumptions and forced conclusion statements. Overall you have statistical differences suggesting a benefit of consecutive motor and cognitive treatment in PD patients in x, y , z outcomes.
Reviewer 2 Report
Review report
Comments to the authors
General comments
In general, the manuscript is easy to read and its subject is interesting. It will contribute to possible improvements of clinical practice. According to the sample size, this should be considered as a pilot study which should be noted in the title, in the methods section and the discussion. In addition, the registration ID of the study is not given in the manuscript. The authors should register the study or if it is already registered they should give details.
Specific comments
The following points should be addressed by the authors.
Title and study design: considering the small sample size, this study should be considered as a pilot study. Please highlight it in the title, design and discussion.
Abstract
Please separate the information in subsections: Background, objective, methods, results and conclusions.
Introduction
Please specify the objective of the study at the end of the introduction section instead of saying the comparison that you made in the study.
Materials and Methods
Please specify if the study has been registered, where and when, and which is the registration ID.
Please re-organise this section including the following subsections and recommended order: study design, ethical aspects, Subjects (target population and inclusion and exclusion criteria), interventions and procedure, data collection and assessment procedure, outcome measures (primary and secondary), statistical analysis.
Interventions: please describe interventions following the ‘Better reporting of interventions: template for intervention description and replication (TIDieR) checklist and guide’. And add a TIDieR table in supplemental material.
Outcomes measures: please specify the outcome measures first and then proceed to explain the assessment tools used.
Please describe how the randomisation was performed.
Discussion
Please re-phrase: ‘Rest on study with other type of patients [33], we supposed that this change translates into clinically meaningful improvement for the patients’. The first part of the sentence does not make sense.
Please explain in the manuscript why you did not used the reviewed version of the UPDRS by Movement Disorders Society (MDS-UPDRS).
Re-write: The future studies should be included this version to confirm our data on UPDRS III. As it is grammatically incorrect.
Please extend on the limitations of the study. Explain why the sample size was so small as this is an important issue. Please explain why no follow up measurements were done. Please explain why there is no comparison to a control group which is a very important issue as well.
Please expand on the clinical implications of the results of the study.
Conclusion
Please include a conclusion section where conclusions of the study are easily found.
Author Response
Please see the attachment

This manuscript is a resubmission of an earlier submission. The following is a list of the peer review reports and author responses from that submission.
Round 1
Reviewer 1 Report
Dear Srs.
I carefully read this relevant study and I have some comments:
comment 1 - there is typo error in the title (with)
comment 2 - Given the current growth (and popularity) of dual task cognitive-motor training, it is very important to clarify the protocol is not dual task but consecutive training of cognitive and then motor. The title can be misleading. "Physical versus combined physical and cognitive treatment in patients with Parkinson’s disease: a randomized cross-over study."
Would be clearer: Physical versus consecutive physical and cognitive treatment in people with Parkinson’s disease: a randomized cross-over study.
More accurately would be: Physiotherapy versus consecutive physiotherapy and cognitive treatment in people with Parkinson’s disease: a randomized cross-over study.
Physiotherapy is not purely physical exercise, just understanding the instructions, environment and progressions, learing strategies for movement, talking while moving, etc is cognitive challenges for the patients. It's incorrect to describe it's purely physical.
comment 3 - line 53 - Please revise this sentence; "To our knowledge, there are no studies on the effects of cognitive training alone on motor dysfunctions."There are studies showing the benefits of cognitive training in PD. There are studies showing benefit cognitive training to improve freezing of gait (Walton, C.C., Mowszowski, L., Gilat, M. et al. Cognitive training for freezing of gait in Parkinson’s disease: a randomized controlled trial. npj Parkinson's Disease 4, 15 (2018)).
comment 4 - Revise this sentence: " Furthermore, to our knowledge, no study has investigated clearly the effectiveness of combined physical and cognitive training on motor deficits". Key reference seem to be missing that show that there is evidence: Strouwen, C., Molenaar, E. A. L. M., Münks, L., Keus, S. H. J., Zijlmans, J. C. M., Vandenberghe, W., … Nieuwboer, A. (2017). Training dual tasks together or apart in Parkinson’s disease: Results from the DUALITY trial. Movement Disorders, 32(8), 1201–1210.
comment 5 - -The word physiotherapy and physical therapy is used unchangeable. Please choose one.
comment 6 - The Movement Disorder Society PD diagnostic criteria has become the gold standard in clinical practice and research settings (Postuma et al, Mov. Disord. 2015). The clinical studies should apply the MDS criteria instead of UK PD brain bank criteria.
comment 7 - What was the rationale behind excluding someone with “on-off” fluctuations"? severe unpredictable on off fluctuations that would compromise persons participation in the program is understandable but only on-off fluctuations seem extreme given its a study that included patients in stage 3.
comment 8- The Unified Parkinson’s Disease Rating Scale (UPDRS) originally developed in the 1980s was reviewed in 2008 by Movement disorders society (MDS) and since then the recommendation is that the most current version is preferably used (MDS-UPDRS).
comment 9- Cross-over study designs in PD will have natural progression of the disease as a bias. How much would people be expected to get worse given the progression rate of the disease vrs the benefits of study?
Later in the discuss, this worsening is mentioned but related to disperse assumption.
" However, our work, differently from others [11, 33], found a decrease in global motorperformance (as measured with UPDRS-III) after both training. We supposed that the UPDRS-III is a global measure that investigated all movement dysfunctions in patients with PD, while our physical training focusing on walking and balance."
comment 10- A clear description of what the interventions consisted of should be included, even if in appendix. It makes it replicable to others and also:
"No drop-outs or adverse events were recorded during the study" - can only be easily accepted knowing the intervention components
comment 11 - The MoCA is recommended to be assessed inside a time-frame in clinical practise. Is this time-frame corresponding to the assessed times in this study? The alternative/equivalent versions (7.2 and 7.3 versions) of the MoCA should be used to decrease possible learning effects when the MoCA is administered repetitively, for example, every 3 months or less.
comment 12 -line 228 again revise this statement: To the best of our knowledge, no studies to date have investigated the effects of combined cognitive-motor intervention on functional mobility.
comment 13 - The discussion is very dense, not clear and objective. Results are justified based on studies that studied dual task and yet this is not dual task training. Very confusing to follow. Too many assumptions. Revision to simply would be warranted.
comment 14 - Please minimise the use of " to our knowledge," too excessive and risky.
Reviewer 2 Report
Varalta et cols present a work aimed at exploring the effect of isolated physical therapy vs physical therapy plus cognitive therapy in patients with Parkinson's disease.
The subject is undoubtedly interesting for this field of study, especially if we take into account that we do not have really effective treatments for cognitive impairment in this disease. Despite this, there are a series of elements that, in my opinion, discourage acceptance of the manuscript in its current form.
Trying to be as concrete as possible, I consider that the findings, although positive to a certain extent, do not allow for broad generalizations nor can they be seen as revealing large effects mediated by the intervention. In this sense, I think it is necessary to limit the tone of the conclusions to the observed reality.
At the same time, there are a series of elements related to the statement of the state of the question and the development of the hypotheses that I consider deserve to be developed in greater depth.
Specifically:
- The first sentence of the abstract states that the cognitive and physical problems of Parkinson's disease (PD) can be effectively addressed through physical therapy and cognitive training. This is unfortunately not the case. There are studies that show the existence of certain benefits (between minimal and partial) but no study has shown that physiotherapy or cognitive training associates great improvements.
- In the introduction, the description of the clinical (motor) characteristics of PD should follow the classic description of the cardinal symptoms of the disease (that is, resting tremor, bradykinesia, postural instability).
- It is one thing for non-motor symptoms to play a fundamental role in the quality of life of patients, and it is quite another, as stated, for cognitive symptoms to play a transcendent role in motor symptoms.
- Similarly, studies that have found a relationship between cognitive and motor improvement do not necessarily demonstrate or corroborate the cognitive-motor relationship. In any case, the authors should develop more and better what they refer to when they speak of the cognitive-motor relationship.
- The paragraph on line 59 that refers to "Administration of physiotherapy and cognitive training in isolation can improve cognition ..." is not spelled correctly.
- It would be interesting to know what methodology was used for the evaluation of the elements that make up the exclusion criteria described.
- It is absolutely necessary to present in detail what the physical therapy program and the cognitive stimulation program consisted of. What elements and domains were worked and based on what, it was decided to work these elements.
- The use of compensation mechanisms as a strategy to mitigate cognitive problems is useful in neuropsychological rehabilitation, but generally, cognitive stimulation does not seek this but rather seeks to stimulate or improve cognitive functioning per se. Why did they opt for compensation strategies? What did the sessions consist of?
- The part B of the TMT is not a measure of attention. It is a measure of cognitive flexibility.
- Before making any comparison, the minimum significant clinical change should have been estimated from the sample. The existence of differences in X scores does not necessarily mean that there is a clinical benefit.
- Based on Table 1, it is somewhat contradictory that patients with a UPDRS of 3 are on a Hoehn & Yahr stage of 3 ... could you explain this?
- I understand that the analysis performed has simply been to make a comparison between groups of baseline scores and a comparison between groups of post-treatment scores. It is not the correct statistical approximation. In these cases, it is ideal to use repeated measures ANOVA and, depending on the interactions or effects, make post-hoc comparisons.
- Regardless of the p values, we cannot expect and affirm that these differences translate into a clinical change or functional improvement since they are minimal. Possibly the effect size is too low. The discussion should adjust to this reality and not speculate with great findings that do not exist.
- Taking into account that no significant results are obtained in the primary outcompe, the discussion should reflect this reality.